# A Graph Theoretic Approach to Construct Desired Cryptographic Boolean Functions

**Modjtaba Ghorbani** [1,*]**, Matthias Dehmer** [2,3,4]**, Vahid Taghvayi-Yazdelli** [1]
**and Frank Emmert-Streib** [5,6]

1  Department of Mathematics, Faculty of Science, Shahid Rajaee, Teacher Training University, Tehran 16785-136, Iran; v.taghvayi@gmail.com
2  Steyr School of Management, University of Applied Sciences Upper Austria, 4400 Steyr, Austria; matthias.dehmer@fh-steyr.at
3  Steyr School of Management, University of Applied Sciences Upper Austria, 4400 Steyr, Austria
4  College of Artificial Intelligence, Nankai University, Tianjin 300071, China
5  Predictive Medicine and Data Analytics Lab, Department of Signal Processing, Tampere University of Technology, 33100 Tampere, Finland; frank.emmert-streib@tut.fi
6  Predictive Society and Data Analytics Lab, Faculty of Information Technology and Communication Sciences, Tampere University, 33720 Tampere, Finland
*  Correspondence: mghorbani@sru.ac.ir

**Abstract:** In this paper, we present four product operations to construct cryptographic boolean functions from smaller ones with predictable Walsh spectrum. A lot of cryptographic properties of boolean functions can be presented by their Walsh spectrum. In our method, we use the product of Cayley graphs to present new boolean functions with desired Walsh spectrum and investigate their non-linearity, algebraic and correlation immunity.

**Keywords:** boolean functions; Walsh spectrum; Cayley graphs; algebraic immunity; non-linearity

## 1. Introduction

Boolean functions are fundamental components of a cryptographic algorithm. Designing boolean functions with desired cryptographic properties is an important problem. Boolean functions should have some properties like balancedness, high non-linearity, algebraic immunity, correlation immunity and propagation criterion to be used in a symmetric algorithm. These properties make the cipher resistant against attacks like differential and linear cryptanalysis, correlation and algebraic attacks and statistical tests. Correlation immune functions were introduced by Siegenthaler [1] to resist against a class of divide and conquer attacks on certain models of stream ciphers. Algebraic attacks [2,3] have become a powerful tool that can be used for almost all types of cryptographic systems. Algebraic attacks will be more efficient if boolean functions have low degrees. All cryptographic properties can be measured by the Walsh spectrum of a boolean function. Hence, constructing boolean functions with desired Walsh spectrum can help designers to use practical boolean functions. In this paper, we aim to propose four new methods to construct larger boolean functions from smaller ones with predictable cryptographic properties.

In 1999 Bernasconi et al. made a new characterization for boolean functions exploiting the graph theoretic approach. They proved that any boolean function can be represented by a Cayley graph and eigenvalues of this Cayley graph correspond to its Walsh spectrum. The authors defined a Cayley graph on the boolean function as $f : \mathcal{Z}_2^n \longrightarrow \mathcal{Z}_2$, where $\Gamma = Cay(\mathcal{Z}_2^n, \Omega_f)$ where $\Omega_f$ is the support of $f$ and two vertices $x, y \in \mathcal{Z}_2^n$ are incident if $x \oplus y \in \Omega_f$. Since for all elements $x \in \mathcal{Z}_2^n$ we have $x^{-1} = x$, any subset of $\mathcal{Z}_2^n$ is symmetric. The adjacency matrix of a boolean function $A_f$ is the adjacency matrix

of its associated Cayley graph and $(A_f)_{ij} = f(i \oplus j)$. The following theorem is the main result of [4] where the function $W(f)$ is the Walsh transform of the boolean function $f$.

**Theorem 1.** *Let $f : \mathcal{Z}_2^n \longrightarrow \mathcal{Z}_2$, and let $\lambda_i, 0 \leq i \leq 2^n - 1$ be the eigenvalues of its associated Cayley graph.*
*(i) For any $0 \leq i \leq 2^n - 1, \lambda_i = W(f)(i)$.*
*(ii) The multiplicity of the largest spectral coefficient of $f$, $W(f)(0)$ is equal to $2^{n - dim\langle \Omega_f \rangle}$.*

In [5] Stanica investigated some cryptographic properties of boolean functions in terms of their eigenvalues and adapted some graph theory concepts with cryptographic properties of these functions. In the following we state some of these results.

**Corollary 1.** *Let $f$ be a boolean function and the eigenvalues of $\Gamma_f$ be ordered as $\lambda_1 \geq \lambda_2 \geq \cdots \geq \lambda_v$.*

1. *Let $g$ be the multiplicity of the lowest eigenvalue of $\Gamma_f$ and $\lambda_2 \neq 0$, in this case $min\{g + 1, 1 - \frac{\lambda_v}{\lambda_2}\} \leq \chi(\Gamma_f) \leq |\Omega_f|$ where $\chi(\Gamma_f)$ is the chromatic number of $\Gamma_f$.*
2. *A boolean function $f$ depends linearly on a variable $x_i$ if and only if the eigenvalues for Cayley graph $\Gamma_f$ satisfy $\lambda_0 = 2^{n-1}$ and if $i$-th component of binary representation of $a \neq 0$ equals to 0, then $\lambda_a = 0$.*
3. *For an unbalanced correlation immune function $f$ on $\mathcal{Z}_2^n$ of order $l$, there are $\sum_{s=1}^{l} \binom{2^n}{s}$ zero eigenvalues of $\Gamma_f$.*

In Section 2, we first start with some definitions and basic properties of boolean functions and their cryptographic properties. Next we state some algebraic graph theory that will be used throughout the paper. In Section 3, we first present some related works to our subject and next we propose our methods to construct new boolean functions. Finally, we present some theorems to investigate their cryptographic properties.

## 2. Definitions and Preliminaries

### 2.1. Boolean Functions

Consider the field $\mathcal{Z}_2$ with elements $\{0, 1\}$. Let $n$ be a positive integer and $\mathcal{Z}_2^n = \{(x_n, ..., x_1) | x_i \in \mathcal{Z}_2, 1 \leq i \leq n\}$ be the binary representation of the positive integer set $\{0, 1, ..., 2^n - 1\}$. Here, we show addition in integer sets by $+$, and addition by module 2 by $\oplus$. For $x, y \in \mathcal{Z}_2^n$, define $x \oplus y = (x_1 \oplus y_1, ..., x_n \oplus y_n)$ and $x.y = x_1 y_1 \oplus ... \oplus x_n y_n$. We also define the Hamming weight of a vector $x \in \mathcal{Z}_2^n$ as $hwt(x) = |\{i | x_i = 1\}|$. Now an $n$-variable boolean function $f(x_n, ..., x_2, x_1)$ is a map from $\mathcal{Z}_2^n$ to $\mathcal{Z}_2$. The set of all boolean functions over $\mathcal{Z}_2^n$ is denoted by $F_n$. An $n$-variable boolean function $f(x_n, ..., x_2, x_1)$ can be represented as a multivariate polynomial over $\mathcal{Z}_2$ called algebraic normal form (ANF) as

$$f(x_n, ..., x_1) = a_0 \bigoplus_{i=0}^{n} a_i x_i \bigoplus_{1 \leq i,j \leq n} a_{ij} x_i x_j \bigoplus \cdots \bigoplus a_{1...n} x_1 x_2 \cdots x_n.$$

The algebraic degree of a boolean function $f$ is the number of variables in the most existing variables of its multiplicative terms with nonzero coefficient in its ANF and denoted by $deg(f)$. A boolean function of the form $f(x) = a_0 \oplus a_1 x_1 \oplus ... \oplus a_n x_n$ is called affine and if $a_0 = 0$, is called linear.

The binary vector of length $2^n$ as $(f(0), f(1), ..., f(2^n - 1)) \in \mathcal{Z}_2^n$, is said to be the truth table of $f$. We also denote the support of $f$ by $\Omega_f$ as $\{x \in \mathcal{Z}_2^n | f(x) = 1\}$. Clearly $0 \leq |\Omega_f| \leq 2^n$ and also the hamming weight of $f$ is $hwt(f) = |\Omega_f|$.

One can see that the ANF of a boolean function $f : \mathcal{Z}_2^n \longrightarrow \mathcal{Z}_2$ with support $\Omega_f$ is

$$f(x_n, \cdots, x_1) = \bigoplus_{\alpha = (\alpha_n, \cdots, \alpha_1) \in \Omega_f} x^\alpha$$

where $x^\alpha = x_n^{(\alpha_n)} \cdots x_1^{(\alpha_1)} = (x_n \oplus \alpha_n \oplus 1) \cdots (x_1 \oplus \alpha_1 \oplus 1)$. So we have

$$f(x_n, \cdots, x_1) = \bigoplus_{\alpha \in \Omega_f} \prod_{i=1}^{n} (x_i \oplus \alpha_i \oplus 1). \tag{1}$$

The distance between two boolean functions $f, g \in F_n$ is defined as $d(f, g) = hwt(f \oplus g)$. For each boolean function $f : \mathcal{Z}_2^n \longrightarrow \mathcal{Z}_2$, the sign function is $\widehat{f} : \mathcal{Z}_2^n \longrightarrow \{-1, 1\}$ where $\widehat{f}(x) = (-1)^{f(x)}$.

The Walsh transform of a boolean function is a function $W(f)$ from $\mathcal{Z}_2^n$ to $\mathcal{Z}$ where

$$W(f)(w) = \sum_{x=0}^{2^n-1} f(x)(-1)^{w.x}.$$

It is clear that $W(f)(0) = |\Omega_f|$ and the Walsh spectrum of a boolean function can be defined as $(W(f)(0), \cdots, W(f)(2^n - 1))$. Also the relation between $W(f)(w)$ and $W(\widehat{f})(w) = \sum_{x=0}^{2^n-1} (-1)^{f(x)+w.x}$ is as follows

$$W(f)(w) = 2^{n-1} \delta(w) - \frac{1}{2} W(\widehat{f})(w), \tag{2}$$

$$W(\widehat{f})(w) = 2^n \delta(w) - 2W(f)(w), \tag{3}$$

where $\delta(w) = 1$ if $w = 0$ and $\delta(w) = 0$, otherwise.

Any cryptographical property comes from a concrete attack or a potential security threat to cryptosystems. A lot of cryptographic properties of boolean functions can be described by their Walsh spectrum. The Walsh spectrum has been mostly employed for analysis and generalization of desired cryptographic boolean functions [6–8].

A boolean function $f$ is called balanced if the number of outputs 0 is equal to the number of outputs 1 or $|\Omega_f| = 2^{n-1}$.

The nonlinearity of a boolean function $f$ denoted by $nl(f)$ is an important criterion to measure the distance of the boolean function with the set of all affine functions. This criterion is a security property to measure how resistant a boolean function is against linear cryptanalysis which is a common and strong attack. If a boolean function has low nonlinearity, an attacker can approximate it by an affine function. A boolean function with high nonlinearity cannot be approximated by linear or affine functions. The nonlinearity of a boolean function can be measured by the following lemma.

**Lemma 1.** *[9] For an n-variable boolean function $f$,*

$$nl(f) = 2^{n-1} - \frac{1}{2} max_{w \in \mathcal{Z}_2^n} |W(\widehat{f})(w)|.$$

By Lemma 1 and Equation (3), one can deduce the following lemma.

**Lemma 2.**
$$nl(f) = min\{hwt(f), 2^n - hwt(f), 2^{n-1} - max_{w \in \mathcal{Z}_2^n} |W(f)(w)|\}.$$

If a boolean function $f$ is a statistically independent of any $k$ of its input variables, then we call $f$ is the correlation immune of order $k$ or $k$-CI which is a security measure about how resistant a boolean function is against a correlation attack. If a boolean function is balanced and correlation immunity is of order $k$, then it is said to be $k$-resilient. Correlation immune (CI) boolean functions were introduced by Siegenthaler [1] to introduce a criterion to resist against certain types of divide and conquer cryptanalysis on some kind of stream ciphers. In 1985, Chor et al. [10] conjectured that the only one-resilient symmetric functions are the exclusive- or of all $n$-variable and its negation. This conjecture was disproved by Gopalakrishnan, Hoffman and Stinson in [11] in 1993 by presenting a class of infinite counter examples, and they noted that it does not seem to extend any further in an

obvious way. Maitra et al. [12] in 1999 proved that the number of $n$-variable CI boolean functions with Hamming weight $2t + 2$ is strictly greater than the number of these functions with Hamming weight $2t$ for which $2t < 2^{n-1}$.

The relation of correlation immunity of a boolean function with its Walsh spectrum can be interpreted as follows.

**Lemma 3.** *A boolean function is correlation immune of order $k$ if and only if for any $w \in \mathcal{Z}_2^n$ with $1 \le hwt(w) \le k$, $W(f)(w) = 0$.*

The algebraic immunity is a cryptographic property that measures the resistance of a boolean function against the algebraic attack proposed by Courtois in 2003 in [2] for stream ciphers and also for block ciphers in [3]. Constructing boolean functions with the best algebraic immunity is a very hard task. The algebraic immunity is an important property of boolean functions that causes resistance against algebraic attack. An algebraic attack is a powerful attack which might threaten the security of both blocks and stream ciphers. The idea is to set up an algebraic system of equations verified by the key bits and trying to solve it. This raises the fundamental issue of determining whether or not a given function has non-trivial low degree multiples. Further, the algebraic immunity of the boolean function $f \in F_n$ is defined as follows:

$$AI(f) = min\{deg(g)|g \in F_n, (fg = 0 \text{ or } (f \oplus 1)g = 0)\}.$$

The problem of finding boolean functions with high algebraic immunity is equivalent to the problem of resisting against algebraic attack. There is a theorem from [13] that characterizes the relation between the support of a boolean function and its algebraic immunity as the following:

**Theorem 2.** *If the support of a boolean function $f$ on $n$ variables is a $k$-dimensional subspace of $\mathcal{Z}_2^n$, then the algebraic degree of $f$ is $n - k$.*

*2.2. Graph Theory*

Since we use some results in graph theory and algebraic graph theory, we now recall some concepts and results in these field in the following.

The set of vertices and edges of a graph $G$ are denoted by $V(G)$ and $E(G)$, respectively. A graph with $n$ vertices and $m$ edges is called $(n, m)$-graph. Two vertices are adjacent if there is an edge between them. A graph is called $r$-regular if every vertex is adjacent to exactly $r$ vertices.

The adjacency matrix of $G$ is a (0,1)-matrix $A = (a_{ij})_{n \times n}$ the rows and columns of which are labeled by vertices and if $v_i v_j \in E(G), (1 \le i, j \le n)$ then $a_{ij} = 1$ and $a_{ij} = 0$, otherwise. The determinant $det(A - \lambda I)$ is a polynomial of degree $n$ which is called characteristic polynomial of $G$. The roots of this polynomial are called the eigenvalues of $G$. If $\lambda_1 > \lambda_2 > ... > \lambda_k$ are distinct eigenvalues of $G$ with multiplicities $s_1, s_2, ..., s_k$ respectively, we denote the spectrum of $G$ by $spec(G) = \{\lambda_1^{s_1}, ..., \lambda_k^{s_k}\}$.

Now we define the Cayley graph from [14]. Let $A$ be a group and $S_A$ be a subset of $A$ such that $1_A \notin S_A$ and if $x \in S_A$ then $x^{-1} \in S_A$(symmetric subset of the group), the Cayley graph $\Gamma = Cay(A, S_A)$ is a simple graph where

$$V(\Gamma) = A \text{ and } E(\Gamma) = \{\{g, h\}|gh^{-1} \in S_A\}.$$

The Cayley graph $\Gamma = Cay(A, S_A)$ is a $|S_A|$-regular graph. Here, we use [15] to introduce four graph products namely, the Cartesian product, tensor product, strong product and lexicographic product and then we investigate some algebraic approachs for the Walsh spectrum of regarded boolean functions. The main question is under which conditions, the product of two Cayley graphs, is again a Cayley graph.

The Cartesian product of two graphs $G_1$ and $G_2$ denoted by $G_1 \square G_2$ is the graph with vertex set $V(G_1) \times V(G_2)$ and edge set

$$\{(v_1, u_1)(v_2, u_2) |\ v_1 = v_2,\ u_1 u_2 \in E(G_2)\ or\ v_1 v_2 \in E(G_1),\ u_1 = u_2\}.$$

If $\{\alpha_i | 1 \le i \le n_1\}$ and $\{\beta_j | 1 \le j \le n_2\}$ are the eigenvalue multisets of $G_1$ and $G_2$ (respectively), then $\{\alpha_i + \beta_j, |1 \le i \le n_1, 1 \le j \le n_2\}$ is the eigenvalue multiset of $G_1 \square G_2$. In the following theorem, the conditions which shows when a Cartesian product of two Cayley graphs is again a Cayley graph, is investigated.

**Theorem 3.** *[16] Let A and B be two groups. If $\Gamma_1 = Cay(A, S_A)$ and $\Gamma_2 = Cay(B, S_B)$ are two Cayley graphs, then the Cartesian product of two graphs $\Gamma = \Gamma_1 \square \Gamma_2$ is also a Cayley graph and $\Gamma = Cay(A \times B, S)$ where $S = (S_A, 1_B) \cup (1_A, S_B)$.*

Let $G_1$ be an $(n_1, m_1)$-graph and $G_2$ an $(n_2, m_2)$-graph. The tensor product $G_1 \otimes G_2$ is a graph with vertex set $V(G_1) \times V(G_2)$ and $E(G_1 \otimes G_2) = \{(v_1, u_1)(v_2, u_2) |\ v_1 v_2 \in E(G_1),\ u_1 u_2 \in E(G_2)\}$. Suppose $\{\alpha_i | 1 \le i \le n_1\}$ and $\{\beta_j | 1 \le j \le n_2\}$ are respectively the eigenvalue multisets of $G_1, G_2$, then $\{\alpha_i \beta_j | 1 \le i \le n_1, 1 \le j \le n_2\}$ is the eigenvalue multiset of $G_1 \otimes G_2$. More generally, we have the following theorem.

**Theorem 4.** *[16] Let A and B be two groups whose related Cayley graphs are $\Gamma_1 = Cay(A, S_A)$ and $\Gamma_2 = Cay(B, S_B)$. Then the tensor product $\Gamma = \Gamma_1 \otimes \Gamma_2$ is also a Cayley graph where $\Gamma = Cay(A \times B, S)$ and $S = S_A \times S_B$.*

The strong product of two graphs $G_1$ and $G_2$ denoted by $G_1 \odot G_2$ is a graph with vertex set $V(G_1) \times V(G_2)$ and edge set $E(G_1 \odot G_2) = E(G_1 \square G_2) \cup E(G_1 \otimes G_2)$. If $\{\alpha_i | 1 \le i \le n_1\}$ and $\{\beta_j | 1 \le j \le n_2\}$ are respectively the eigenvalue multisets of $G_1$ and $G_2$, then $\{(\alpha_i + 1)(\beta_j + 1) - 1, | 1 \le i \le n_1, 1 \le j \le n_2\}$ is the eigenvalue multiset of $G_1 \odot G_2$.

**Theorem 5.** *[16] Let A and B be two groups. If $\Gamma_1 = Cay(A, S_A)$ and $\Gamma_2 = Cay(B, S_B)$ are two Cayley graphs, then the strong product $\Gamma = \Gamma_1 \odot \Gamma_2$ is also a Cayley graph and $\Gamma = Cay(A \times B, S)$, where $S = (S_A \times S_B) \cup (S_A, 1_B) \cup (1_A, S_B)$.*

The lexicographic product $G_1 \circ G_2$ of two graphs $G_1$ and $G_2$ is one with vertex set $V(G_1) \times V(G_2)$ and

$$E(G_1 \circ G_2) = \{(v_1, u_1)(v_2, u_2) | v_1 v_2 \in E(G_1)\ or\ v_1 = v_2,\ u_1 u_2 \in E(G_2)\}.$$

**Theorem 6.** *[17] Let $G_1$ be a graph of order $n_1$ with spectrum $spec(G_1) = \{\lambda_1^{m_1}, \lambda_2^{m_2}, ..., \lambda_s^{m_s}\}$ and let $G_2$ be a p-regular graph of order $n_2$ with spectrum $spec(G_2) = \{\mu_1^{r_1}, \mu_2^{r_2}, ..., \mu_t^{r_t}\}$. Then*

$$\begin{aligned} spec(G_1 \circ G_2) &= \{p^{n_1(r_1-1)}, \mu_2^{n_1 r_2}, \cdots, \mu_t^{n_1 r_t}\} \\ &\cup \{(n_2 \lambda_1 + \mu_1)^{m_1}, \cdots, (n_2 \lambda_s + \mu_1)^{m_s}\}. \end{aligned}$$

**Theorem 7.** *[16] Let A and B be two groups. If $\Gamma_1 = Cay(A, S_A)$ and $\Gamma_2 = Cay(B, S_B)$ are two Cayley graphs, then the lexicographic product $\Gamma = \Gamma_1 \circ \Gamma_2$ is also a Cayley graph in which $\Gamma = Cay(A \times B, S)$ and $S = (S_A \times B) \cup (1_A, S_B)$.*

## 3. Main Results

Constructing cryptographic boolean functions with different methods is a prevalent research field.

Here, we construct four new families of boolean functions by means of graph products with predictable Walsh spectrum and then we verify their correlation immunity, algebraic immunity and nonlinearity. For a given boolean function $f$, we suppose $f(0) = 0$ or the associated Cayley graph is simple. Let $f_1 : \mathcal{Z}_2^{n_1} \longrightarrow \mathcal{Z}_2$ and $f_2 : \mathcal{Z}_2^{n_2} \longrightarrow \mathcal{Z}_2$ be two boolean functions with associated Cayley graphs $\Gamma_1 = Cay(\mathcal{Z}_2^{n_1}, \Omega_{f_1})$ and $\Gamma_2 = Cay(\mathcal{Z}_2^{n_2}, \Omega_{f_2})$, respectively. In the following, suppose $[n] = \{1, 2, \cdots, n\}$.

First construction. The Cartesian product of two boolean functions can be defined as $f_1 \square f_2 : \mathcal{Z}_2^{n_1+n_2} \longrightarrow \mathcal{Z}_2$ with

$$
\begin{aligned}
\Omega_{f_1 \square f_2} &= \{x \in \mathcal{Z}_2^{n_1+n_2} | (x_{n_1+n_2}, \cdots, x_{n_1+1}) \in \Omega_{f_2}, \ x_i = 0, i \in [n_1]\} \\
&\cup \{x \in \mathcal{Z}_2^{n_1+n_2} | (x_{n_1}, \cdots, x_1) \in \Omega_{f_1}, \ x_{i+n_1} = 0, i \in [n_2]\}.
\end{aligned}
$$

Clearly, $|\Omega_{f_1 \square f_2}| = |\Omega_{f_1}| + |\Omega_{f_2}|$ and by Equation (1), one can conclude that the ANF of $f = f_1 \square f_2$ is

$$
\begin{aligned}
f(x_{n_1+n_2}, \cdots, x_2, x_1) &= f_2(x_{n_1+n_2}, \cdots, x_{n_1+1}) \square f_1(x_{n_1}, \cdots, x_1) \\
&= (1 \oplus x_{n_1+n_2}) \cdots (1 \oplus x_{n_1+1}) f_1(x_{n_1}, \cdots, x_1) \\
&\oplus (1 \oplus x_{n_1}) \cdots (1 \oplus x_1) f_2(x_{n_1+n_2}, \cdots, x_{n_1+1}).
\end{aligned}
$$

The spectrum of the related Cayley graph on $f_1 \square f_2$ is

$$
\{W(f_1)(x) + W(f_2)(y) | x \in \mathcal{Z}_2^{n_1}, y \in \mathcal{Z}_2^{n_2}\}.
$$

Second construction. The tensor product of two boolean functions can be defined as $f_1 \otimes f_2 : \mathcal{Z}_2^{n_1+n_2} \longrightarrow \mathcal{Z}_2$ with

$$
\Omega_{f_1 \otimes f_2} = \{x \in \mathcal{Z}_2^{n_1+n_2} | (x_{n_1+n_2}, ..., x_{n_1+1}) \in \Omega_{f_2}, \ (x_{n_1}, \cdots, x_1) \in \Omega_{f_1}\}.
$$

One can conclude that the ANF of $f = f_1 \otimes f_2$ is

$$
\begin{aligned}
f(x_{n_1+n_2}, \cdots, x_1) &= f_2(x_{n_1+n_2}, \cdots, x_{n_1+1}) \otimes f_1(x_{n_1}, \cdots, x_1) \\
&= f_2(x_{n_1+n_2}, \cdots, x_{n_1+1}) f_1(x_{n_1}, \cdots, x_1),
\end{aligned}
$$

where, on the other hand, the spectrum of the related Cayley graph on $f_1 \otimes f_2$ is

$$
\{W(f_1)(x) W(f_2)(y) | x \in \mathcal{Z}_2^{n_1}, y \in \mathcal{Z}_2^{n_2}\}.
$$

Third construction. In this construction we define the strong product of two boolean functions as $f = f_1 \odot f_2 : \mathcal{Z}_2^{n_1+n_2} \longrightarrow \mathcal{Z}_2$ with

$$
\begin{aligned}
\Omega_f &= \{x \in \mathcal{Z}_2^{n_1+n_2} | (x_{n_1+n_2}, \cdots, x_{n_1+1}) \in \Omega_{f_2}, (x_{n_1}, \cdots, x_1) \in \Omega_{f_1}\} \\
&\cup \{x \in \mathcal{Z}_2^{n_1+n_2} | (x_{n_1+n_2}, \cdots, x_{n_1+1}) \in \Omega_{f_2}, \ x_i = 0, i \in [n_1]\} \\
&\cup \{x \in \mathcal{Z}_2^{n_1+n_2} | (x_{n_1}, \cdots, x_1) \in \Omega_{f_1}, \ x_{i+n_1} = 0, i \in [n_2]\}.
\end{aligned}
$$

By Equation (1), it is easy to verify that the ANF of $f = f_1 \odot f_2$ is

$$
\begin{aligned}
f(x_{n_1+n_2}, \cdots, x_1) &= f_2(x_{n_1+n_2}, \cdots, x_{n_1+1}) \odot f_1(x_{n_1}, \cdots, x_1) \\
&= f_2(x_{n_1+n_2}, \cdots, x_{n_1+1}) f_1(x_{n_1}, \cdots, x_1) \\
&\oplus (1 \oplus x_{n_1+n_2}) \cdots (1 \oplus x_{n_1+1}) f_1(x_{n_1}, \cdots, x_1) \\
&\oplus (1 \oplus x_{n_1}) \cdots (1 \oplus x_1) f_2(x_{n_1+n_2}, \cdots, x_{n_1+1}),
\end{aligned}
$$

where $|\Omega_{f_1 \odot f_2}| = |\Omega_{f_1}||\Omega_{f_2}| + |\Omega_{f_1}| + |\Omega_{f_2}|$. The spectrum of the related Cayley graph on $f_1 \odot f_2$ is

$$\{W(f_1)(x)W(f_2)(y) + W(f_1)(x) + W(f_2)(y)|x \in \mathcal{Z}_2^{n_1}, y \in \mathcal{Z}_2^{n_2}\}.$$

Fourth construction. Here we define the lexicographic product of two boolean functions as $f = f_1 \circ f_2 : \mathcal{Z}_2^{n_1+n_2} \longrightarrow \mathcal{Z}_2$ with

$$\begin{aligned}
\Omega_f \quad &= \quad \{x \in \mathcal{Z}_2^{n_1+n_2}|(x_{n_1+n_2}, \cdots, x_{n_1+1}) \in \mathcal{Z}_2^{n_2} \text{ and } (x_{n_1}, \cdots, x_1) \in \Omega_{f_1}\} \\
&\cup \quad \{x \in \mathcal{Z}_2^{n_1+n_2}|(x_{n_1+n_2}, \cdots, x_{n_1+1}) \in \Omega_{f_2} \text{ and } x_i = 0, 1 \le i \le n_1\}.
\end{aligned}$$

where $|\Omega_{f_1 \circ f_2}| = 2^{n_2}|\Omega_{f_1}| + |\Omega_{f_2}|$. By Equation (1), one can see that the ANF of $f = f_1 \circ f_2$ is

$$\begin{aligned}
f(x_{n_1+n_2}, \cdots, x_1) \quad &= \quad f_2(x_{n_1+n_2}, \cdots, x_{n_1+1}) \circ f_1(x_{n_1}, \cdots, x_1) \\
&= \quad [f_1(x_{n_1}, \cdots, x_1) \oplus (1 \oplus x_{n_1})] \cdots \\
&\quad\quad [(1 \oplus x_1)f_2(x_{n_1+n_2}, \cdots, x_{n_1+1})].
\end{aligned}$$

The spectrum of the related Cayley graph on $f_1 \circ f_2$ is

$$spec(f_2) \cup \{2^{n_2}W(f_1)(x) + W(f_2)(0)|x \in \mathcal{Z}_2^{n_1}\}.$$

Now we verify the cryptographic properties of these constructions. In the following theorems, let $f_1$ and $f_2$ be as above.

**Theorem 8.** *In construction one, if $f_1$ and $f_2$ are correlation immune of order $l$, then $f_1 \square f_2$ is correlation immune of order $l$. Also if $f_1$ and $f_2$ are correlation immune of order $l$ such that $|\Omega_{f_1}| + |\Omega_{f_2}| = 2^{n_1+n_2-1}$, then $f_1 \square f_2$ is $l-$resilient.*

**Proof.** Let $f = f_1 \square f_2$ and $w = (w_{n_1+n_2}, ..., w_1) \in \mathcal{Z}_2^{n_1+n_2}$ in which $hwt(w) = l$. Let $w = (a, b), a \in \mathcal{Z}_2^{n_2}, b \in \mathcal{Z}_2^{n_1}$, then

$$\begin{aligned}
W(f)(w) \quad &= \quad \sum_{x \in \mathcal{Z}_2^{n_1+n_2}, x=(r,s)} f(x)(-1)^{w.x} = \sum_{x \in \Omega_{f_1 \square f_2}} (-1)^{w.x} \\
&= \quad \sum_{r \in \Omega_{f_2}, s=0} (-1)^{w.x} + \sum_{r=0, s \in \Omega_{f_1}} (-1)^{w.x} \\
&= \quad \sum_{r \in \Omega_{f_2}} (-1)^{a.r} + \sum_{s \in \Omega_{f_1}} (-1)^{b.s} \\
&= \quad W(f_2)(a) + W(f_1)(b).
\end{aligned}$$

Since $hwt(a), hwt(b) \le l$ and $f_1$ and $f_2$ are $l-$CI, we can verify that $W(f_2)(a) = 0, W(f_1)(b) = 0$ and hence $W(f)(w) = 0$. $\square$

**Theorem 9.** *In the second construction, if $f_1$ and $f_2$ are $l$-CI, then $f_1 \otimes f_2$ is $l$-CI, and if $|\Omega_{f_1}||\Omega_{f_2}| = 2^{n_1+n_2-1}$, then $f_1 \otimes f_2$ is $l$-resilient.*

**Proof.** For any $w \in \mathcal{Z}_2^{n_1+n_2}$ with $hwt(w) = l$ and $w = (a, b)$, where $a \in \mathcal{Z}_2^{n_2}, b \in \mathcal{Z}_2^{n_1}$, we prove $W(f)(w) = W(f_2)(a)W(f_1)(b)$. This yields that

$$
\begin{aligned}
W(f)(w) &= \sum_{x \in \mathcal{Z}_2^{n_1+n_2}, x=(r,s)} f(x)(-1)^{w.x} \\
&= \sum_{x \in \Omega_{f_1 \otimes f_2}} (-1)^{w.x} \\
&= \sum_{r \in \Omega_{f_2}, s \in \Omega_{f_1}} (-1)^{a.r+b.s} \\
&= \sum_{r \in \Omega_{f_2}, s \in \Omega_{f_1}} (-1)^{a.r}(-1)^{b.s} \\
&= \sum_{r \in \Omega_{f_2}} (-1)^{a.r} \sum_{s \in \Omega_{f_1}} (-1)^{b.s} \\
&= W(f_2)(a)W(f_1)(b).
\end{aligned}
$$

Since $hwt(a), hwt(b) \leq l$ and $f_1$ and $f_2$ are $l-$CI, we have $W(f_2)(a) = 0, W(f_1)(b) = 0$ and hence $W(f)(w) = 0$. □

**Theorem 10.** *If $f_1$ and $f_2$ are l-CI, then $f_1 \odot f_2$ is l-CI, and if $|\Omega_{f_1}||\Omega_{f_2}| + |\Omega_{f_1}| + |\Omega_{f_2}| = 2^{n_1+n_2-1}$, then $f_1 \odot f_2$ is l-resilient.*

**Proof.** Since $\Omega_{f_1 \odot f_2} = \Omega_{f_1 \square f_2} \cup \Omega_{f_1 \otimes f_2}$, then for any $w \in \mathcal{Z}_2^{n_1+n_2}$ with $hwt(w) = l$ and $w = (a, b), a \in \mathcal{Z}_2^{n_2}, b \in \mathcal{Z}_2^{n_1}$, we have $W(f)(w) = W(f_2)(a) + W(f_1)(b) + W(f_2)(a)W(f_1)(b)$. On the other hand, $f_1, f_2$ are $l-$CI and $hwt(a), hwt(b) \leq l$ which yields that $W(f_2)(a) = 0$ and $W(f_1)(b) = 0$. This means that $W(f)(w) = 0$. □

**Theorem 11.** *For two boolean functions $f_1, f_2$, the boolean function $f_1 \circ f_2$ is not correlation immune.*

**Proof.** Consider the element $w \in \mathcal{Z}_2^{n_1+n_2}$ such that $hwt(w) = l$ and $w = (a, b)$, where $a \in \mathcal{Z}_2^{n_2}, b \in \mathcal{Z}_2^{n_1}$. In this case we have

$$
\begin{aligned}
W(f)(w) &= \sum_{x \in \mathcal{Z}_2^{n_1+n_2}, x=(r,s)} f(x)(-1)^{w.x} \\
&= \sum_{x \in \Omega_{f_1 \circ f_2}} (-1)^{w.x} \\
&= \sum_{r \in \mathcal{Z}_2^{n_2}, s \in \Omega_{f_1}} (-1)^{w.x} + \sum_{r \in \Omega_{f_2}, s=0} (-1)^{w.x} \\
&= \sum_{r \in \mathcal{Z}_2^{n_2}, s \in \Omega_{f_1}} (-1)^{a.r}(-1)^{b.s} + \sum_{r \in \Omega_{f_2}} (-1)^{a.r} \\
&= \sum_{r \in \mathcal{Z}_2^{n_2}} (-1)^{a.r} \sum_{s \in \Omega_{f_1}} (-1)^{b.s} + W(f_2)(a).
\end{aligned}
$$

In other words, if $a = 0$, then $W(f)(w) = 2^{n_2} W(f_1)(b) + W(f_2)(0)$ and if $a \neq 0$ then $W(f)(w) = W(f_2)(a)$. □

The algebraic immunity of these constructions is presented in the following:

**Theorem 12.** *Let $f_1 : \mathcal{Z}_2^{n_1} \longrightarrow \mathcal{Z}_2$ and $f_2 : \mathcal{Z}_2^{n_2} \longrightarrow \mathcal{Z}_2$ be two boolean functions. Then*
*(i) $AI(f_1 \square f_2) = 2$,*
*(ii) $AI(f_1 \otimes f_2) \leq min\{AI(f_1), AI(f_2)\},$*

*(iii) $AI(f_1 \odot f_2) \leq min\{AI(f_1), AI(f_2)\} + 1$,*
*(iv) $AI(f_1 \circ f_2) \leq AI(f_1) + 1$.*

**Proof.** (i) $x_1 x_{n_1}$ is an annihilator of $f_1 \square f_2$ so

$$AI(f_1 \square f_2) = 2.$$

(ii) If $g_1$ and $g_2$ are annihilators of $f_1$ and $f_2$ respectively, then $g_1$ and $g_2$ are annihilators of $f_1 \otimes f_2$ too.

(iii) If $g_1$ and $g_2$ are annihilators of $f_1$ and $f_2$ respectively, then $x_{n_1} g_1$ and $x_{n_1+1} g_2$ are annihilators of $f_1 \odot f_2$ too.

(iv) If $g_1$ is an annihilator of $f_1$, then $x_1 g_1$ is an annihilator of $f_1 \circ f_2$.  □

In the following theorem, we investigate nolinearity of these constructions.

**Theorem 13.** *Let $f_1 \in F_{n_1}$ and $f_2 \in F_{n_2}$. Then*

$$
\begin{aligned}
(i)\ nl(f_1 \square f_2) &= min\{|\Omega_{f_1}| + |\Omega_{f_2}|, 2^{n_1+n_2} - |\Omega_{f_1}| + |\Omega_{f_2}|, 2^{n_1+n_2-1} \\
&\quad - max_{(x,y)\neq 0}\{|W(f_1)(x) + W(f_2)(y)|\}\}. \\
(ii)\ nl(f_1 \otimes f_2) &= min\{|\Omega_{f_1}||\Omega_{f_2}|, 2^{n_1+n_2} - |\Omega_{f_1}||\Omega_{f_2}|, 2^{n_1+n_2-1} \\
&\quad - max_{(x,y)\neq 0}\{|W(f_1)(x)W(f_2)(y)|\}\}. \\
(iii)\ nl(f_1 \odot f_2) &= min\{|\Omega_{f_1}| + |\Omega_{f_2}| + |\Omega_{f_1}||\Omega_{f_2}|, 2^{n_1+n_2} \\
&\quad - (|\Omega_{f_1}| + |\Omega_{f_2}| + |\Omega_{f_1}||\Omega_{f_2}|), 2^{n_1+n_2-1} - max_{(x,y)\neq 0} \\
&\quad \{|W(f_1)(x) + W(f_2)(y) + W(f_1)(x)W(f_2)(y)|\}\}. \\
(iv)\ nl(f_1 \circ f_2) &= min\{2^{n_2}|\Omega_{f_1}| + |\Omega_{f_2}|, 2^{n_1+n_2} - 2^{n_2}|\Omega_{f_1}| + |\Omega_{f_2}|, \\
&\quad 2^{n_1+n_2-1} - max\{W(f_2)(x), 2^{n_2}W(f_1)(y) + |\Omega_{f_2}|\}\}.
\end{aligned}
$$

**Proof.** It is a straight result from Lemma 2 and the following conditions
(i) $hwt(f_1 \square f_2) = |\Omega_{f_1} + \Omega_{f_2}|$,
(ii) $hwt(f_1 \otimes f_2) = |\Omega_{f_1}||\Omega_{f_2}|$,
(iii) $hwt(f_1 \odot f_2) = |\Omega_{f_1}||\Omega_{f_2}| + |\Omega_{f_1}| + |\Omega_{f_2}|$,
(iv) $hwt(f_1 \circ f_2) = 2^{n_2}|\Omega_{f_1}| + |\Omega_{f_2}|$.  □

## 4. Conclusions

Boolean functions have many applications in fault-tolerant distributed computing and quantum cryptographic key. The fundamental tool in analysis of cryptographic boolean functions is the Walsh spectrum. In this paper, we introduced four new constructions of cryptographic boolean functions by using Cayley graph products. These boolean functions are constructed from smaller ones and the Walsh spectrum of them can be derived by Walsh spectrum of the smaller ones. Next, we investigated the conditions of correlation, algebraic immunity and nonlinearity of these families by the smaller boolean functions. These conditions help designers to design large boolean functions with desired cryptography properties. In future works, we can apply our method for other graph products.

**Author Contributions:** M.G., M.D., V.T.-Y., F.E.-S. wrote the paper.

**Funding:** Matthias Dehmer thanks the Austrian Science Funds for supporting this work (project P30031).

**Conflicts of Interest:** The authors declare that there is no conflict of interest.

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
