# Peer review of "A Graph Theoretic Approach to Construct Desired Cryptographic Boolean Functions"

_axioms, doi:10.3390/axioms8020040_

Round 1

Reviewer 1 Report

This paper uses product of Cayley graphs to present new Boolean functions with desired Walsh spectrum and investigates the non-linearity, algebraic and correlation immunity the new Boolean functions. This paper is well written and can be accepted. Some comments are as follows:

1. Some remarks should be given to show the novelties of this paper compared with other related papers.

2. Theorems 3.5 and 3.6 are given directly without any proofs or illustrations. I think some illustrations should be given.

Author Response

This paper uses product of Cayley graphs to present new Boolean functions with desired Walsh spectrum and investigates the non-linearity, algebraic and correlation immunity the new Boolean functions. This paper is well written and can be accepted. Some comments are as follows:

1. Some remarks should be given to show the novelties of this paper compared with other related papers.

Responses The main idea is completely new and based on results of paragraph 2 of page 2, page 3 Theorem  and page 7 paragraph 2. This is the first attemp to construct Boolean functions from graph operations. Also, we applied N. Biggs for Cayley graphs structures.

2. Theorems 3.5 and 3.6 are given directly without any proofs or illustrations. I think some illustrations should be given.

Responses The proofs are added.

Reviewer 2 Report

accepted after gramner correction. 

Author Response

The conclusions are supported by the results.

Reviewer 3 Report

Dear authors,

This manuscript is based on graph theory applied on cryptography. The application of this field of mathematics is interesting and important – a prevalent research field as said by the authors.

The manuscript is very well structured and only few spelling errors occur, which will be handled below. The concept recall in the beginning of section 2.2 is an excellent merit.

In many places there needs to be a “the” before the noun, e.g.

Abstract, line 5: before “product”

Introduction, second paragraph, line 2: behind “exploiting”

Page 6, line 8: before “correlation”

Page 7, line 6: before “algebraic” (or “an”)

Page 7, line 7: before “relation” and before “support”

Page 7, line 5 from below: before “Cayley”

Page 11, line 6 from below: before “cryptographic”

Please also consider the following spelling errors or typos:

Page 2, line 1: “components”

Page 2, line 5: Do you really mean “used”?

Page 2, line 7: “correlation”

Page 2, second paragraph: “et al.”, later it’s written correctly.

Page 3, line 7: “are” à “be”

Page 3, line 17: “we first”

Page 6, line 5: “is” à “a”

Page 6, line 6: “against a”

Page 6, line 7: “is of order”

Page 6, line 16: “In 1999, Maitra et al. [20] proved”

Page 6, line 2 from below: “trying to solve it”?

Page 8, line 4: “appoaches”

Page 8, line 13: “when a Cartesian product”

Page 14, Comclusion, line 4: “derived”

Finally, I think the Conclusion is just a Summary of the derivations and needs to be either rewritten or retitled.

In conclusion, I think this manuscript could be acceptable for publication provided the authors undergo a minor revision.

Author Response

This manuscript is based on graph theory applied on cryptography. The application of this field of mathematics is interesting and important – a prevalent research field as said by the authors.

The manuscript is very well structured and only few spelling errors occur, which will be handled below. The concept recall in the beginning of section 2.2 is an excellent merit.

In many places there needs to be a “the” before the noun, e.g.

Abstract, line 5: before “product”

Introduction, second paragraph, line 2: behind “exploiting”

Page 6, line 8: before “correlation”

Page 7, line 6: before “algebraic” (or “an”)

Page 7, line 7: before “relation” and before “support”

Page 7, line 5 from below: before “Cayley”

Page 11, line 6 from below: before “cryptographic”

Please also consider the following spelling errors or typos:

Page 2, line 1: “components”

Page 2, line 5: Do you really mean “used”?

Page 2, line 7: “correlation”

Page 2, second paragraph: “et al.”, later it’s written correctly.

Page 3, line 7: “are” à “be”

Page 3, line 17: “we first”

Page 6, line 5: “is” à “a”

Page 6, line 6: “against a”

Page 6, line 7: “is of order”

Page 6, line 16: “In 1999, Maitra et al. [20] proved”

Page 6, line 2 from below: “trying to solve it”?

Page 8, line 4: “appoaches”

Page 8, line 13: “when a Cartesian product”

Page 14, Comclusion, line 4: “derived”

ResponsesAll of them are corrected and are hiighted in the text.

Finally, I think the Conclusion is just a Summary of the derivations and needs to be either rewritten or retitled.

ResponsesThe Conclusion is rewritten.